# Assessment of Permanent First Molars in Children Aged 7 to 10 Years Old

**DOI:** 10.3390/children10010061

**Published:** 2022-12-27

**Authors:** Gelengul Urvasizoglu, Aybike Bas, Fatma Sarac, Peris Celikel, Fatih Sengul, Sera Derelioglu

**Affiliations:** Department of Pedodontics, Faculty of Dentistry, Ataturk University, Erzurum 25000, Turkey

**Keywords:** caries, DMFT, first permanent molars, epidemiology, tooth brushing, dental visits

## Abstract

Background: Dental caries is a chronic, infectious and preventable disease that is very common around the world. It has been observed that dental caries affect not only the majority of adults but also 60% to 90% of children. Permanent first molars (PFM) are the most commonly decayed teeth observed in children. Aim: The aim of this study is to evaluate the decayed, missing filled teeth (DMFT) scores of PFMs in the early post-eruptive stage, within the scope of the United Nations Agenda for 2030 Sustainable Development Goals, thereby raising awareness for the prevention and treatment of permanent tooth decay. Methods: This descriptive cross-sectional epidemiological study was conducted in Erzurum between the years 2015–2016 by collecting data from children aged 7–10 years (17,208). In addition to the decayed, filled and missing data of the students’ 6-year-molars, their ages, genders, frequencies of both tooth brushing and dental office visits were evaluated. The relationship between the variables was analyzed with chi-square. Result: The present study analyzed the data of a total of 11,457 children, 5704 girls and 5753 boys with a mean age of 8.74 ± 1.18. There was a statistically significant difference between the PFMs 16, 26, 36 and 46 regarding the number of healthy, decayed, missing and restored teeth (*p* < 0.001). Conclusion: In this study, the prevalence of caries in the PFMs of children aged 7–10 years was 15.9% and the mean DMFT was 0.79 ± 1.39. This result showed that PFMs might develop carious lesions and even be lost within three years in the early post-eruptive stage.

## 1. Introduction

Dental caries is a chronic, infectious and preventable disease that is very common around the world [1]. Today, it is a well-known fact that several predisposing factors, such as different dental anatomical features, amount and fluency of salivary secretion, genetic and sociocultural/socioeconomic status, oral hygiene and dietary habits, and the frequency of dental visits play a role in the caries development [2]. It has been observed that dental caries affect not only the majority of adults but also 60% to 90% of children [1].

Permanent first molars (PFMs) are the most commonly decayed teeth, observed in children [3]. PFMs’ deep pits and fissures and incomplete maturation in their early developmental stages, little children’s inability to perform an effective toothbrushing and also parental misconception that PFMs are deciduous are the major factors causing these caries [3,4]. Regarding functionality and growth, PFMs are the most important teeth playing a key role in occlusion [5]. 

Therefore, regular dental checks, preventive dental treatments, and early caries intervention stop the premature loss of PFMs, and also avoid waste of time and financial losses [6].

Epidemiological studies play an important role in the distribution and frequency of oral diseases and are considered extremely important for public health, as they form the basis for determining treatment needs. In these studies, there is a need for epidemiological indices that are comparable, reproducible, easily applicable, valid and reliable [7]. In epidemiological studies of dental caries, the decayed, missing filled teeth (DMFT/dmft) index comes first among the indexes that meet these features. In the DMFT/dmft index, general condition of dental health is expressed by the sum of the number of decayed (D/d), missing (M/m) and filled (F/f) teeth [8]. 

The third element of 17 Sustainable Development Goals 2030, which were discussed by the United Nations in January 2015 and adopted in September 2015, has been declared as “good health and wellbeing” [9]. Similar goals were set in World Dental Federation’s (Fédération Dentaire Internationale—FDI) strategy of “Delivering Optimal Oral Health for All—Vision 2030” [9,10]. In all of these goals, integration of oral health and general health were remarked as a common point. In order to achieve these goals, it is important to determine the current public oral health status. The studies on oral health of the children in Erzurum revealed that early childhood caries (ECC) rate was 76.6%–73.3% [11,12]. A different study examining the PFMs of children with a history of ECC in the region highlighted that 44.4% of the PFMs of children aged 6–9 years were affected by caries [13]. Studies have argued that ECC was a risk factor for the caries formation in the permanent dentition. 

The high prevalence of ECC in Erzurum province indicated that PFMs of children in this population were under the risk of carious lesions when the permanent dentition started. In this regard, it was found to be highly significant to examine their PFMs. The aim of this study is to evaluate the DMFT scores of PFMs in the early post-eruptive stage, within the scope of the United Nations Agenda for 2030 Sustainable Development Goals, thereby raising awareness for the prevention and treatment of permanent tooth decays.

## 2. Materials and Methods

The present descriptive cross-sectional epidemiological study was conducted by Ataturk University’s Faculty of Dentistry, Department of Pediatric Dentistry in Erzurum, Turkey in the 2015–2016 academic year in accordance with the provisions of the Ministry of Health Clinical Research Regulations. A written consent was also obtained from the Faculty of Medicine’s Research Ethics Committee (Decision No: 9/61- Date: 1 December 2022).

The present cross-sectional study was carried out by collecting data from the 1st–4th grades from all schools (304 schools) with students aged 7–10 years (17,208) in Erzurum. However, only 11,457 students (67%) whose PFMs had fully erupted were included in the study. 

The present study was conducted by 3 faculty members, 5 research assistants and senior students at Ataturk University’s Faculty of Dentistry. Examiners were calibrated for examination using the protocol laid down by the WHO (World Health Organisation) [14]. Researchers whose intra and inter-examiner Kappa reliability scores were close to perfect (greater than 0.81) in accordance with Landis and Koch [15] scaling, performed the oral examinations in the study.

School principals were informed that the children should brush their teeth before the oral examination. Information forms containing the treatment needs were sent to the families of the children. After the oral examination and registration processes completed, varnishes containing 5% sodium fluoride (NaF) were applied to the children’s teeth and they received oral health training.

The WHO protocol was taken as a basis for the dental examinations [14]. Dental examinations of the students were carried out in a classroom by using a dental mirror and a WHO ball-ended periodontal screening probe, in a sitting position. All PFMs were examined in a systematic approach, starting from the upper right PFM and proceeding through the lower right one. The teeth with pit and fissure lesions, unmistakable surface cavities, undermined enamels or with apparent softened floors or walls, were coded as “decayed”. Moreover, PFMs with secondary caries or temporary fillings were also coded as “caries”. Missing tooth code was used for extracted PFMs due to caries. Filled tooth code was used for PFMs with permanent restorations and no caries.

In addition to the DMFT index for PFM caries of the students, their ages, genders, frequencies of both tooth brushing and dental office visits, experiences of bleeding gums while brushing were also recorded in the forms created by the researchers. Radiography was not used in the evaluation of the teeth.

Sample size was calculated as 1459 using Epi Info ^TM^ 6 (Centers for Disease Control and Prevention, Atlanta, GA, USA, with 99% confidence interval, 5% standard error, and 44% prevalence). However, the purpose of the present study was to reach all children in this age group living in Erzurum and 11,185 children were accordingly included in the study. Data provided by Erzurum civil registry office were used in determining the sample size and the number of children aged 5–10 years living in Erzurum and its districts was set as 65,136 accordingly. There is no recent data on the prevalence of PFM caries in Erzurum province. However, in the study sample size calculation, frequency was estimated as 44% because of the high ECC prevalence of the children living in Erzurum [11] and because of the PFM decays in 44% of the children affected by ECC [13]. 

In the present study, statistical tests were carried out at a significance level of 0.05 using SPSS 26.0 (IBM, Armonk, NY, USA). The association between the variables was analyzed with chi-square test and post-hoc chi-square test was conducted to find out if there were differences between the groups when a multi-way (larger contingency table) chi-square test was statistically significant. 

## 3. Results

The present study analyzed the data of a total of 11,457 children, 5704 girls and 5753 boys with a mean age of 8.74 ± 1.18. 

Table 1 indicates the status of sound, decayed, missing and filled PFMs by gender. There was a statistically significant difference between the genders regarding the number of sound, decayed, missing and restored PFMs. The higher number of sound PFMs in boys (*p* < 0.001) revealed a statistically significant difference when compared to the higher number of decayed (*p* < 0.001) and restored (*p* = 0.024) PFMs in girls. The mean number of DMFT for PFMs in these patients was 0.79 ± 1.39, and the mean DMFTs were 0.88 ± 1.44 in girls and 0.69 ± 1.34 in boys. 

Table 2 shows the status of sound, decayed, missing and filled PFMs by age. There was a statistically significant difference between the age regarding the number of sound, decayed, missing and restored PFMs (*p* < 0.001). While the ratio of sound PFMs decreases with the increasing age, there was a significant difference due to the increase in the ratio of decayed, missing and restored PFMs (*p* < 0.001). There was a statistically significant difference in the ratio of decayed PFMs to the restored ones depending on age (*p* < 0.001). 

Table 3 indicates the status of sound, decayed, missing and filled PFMs by the tooth number. There was a statistically significant difference between the PFMs 16, 26, 36 and 46 regarding the number of sound, decayed, missing and restored PFMs (*p* < 0.001). Only the PFMs in the same jaw (maxilla or mandible) showed a similar distribution regarding the number of sound, decayed, missing and restored PFMs (*p* > 0.05). Compared to the high number of sound PFMs in the maxilla, high numbers of decayed (*p* < 0.001) and restored (*p* < 0.001) PFMs in the mandible created a statistically significant difference. There was a statistically significant difference due to the f/d ratio of the maxillary PFMs, which was almost twice as the rate of mandibular PFMs (*p* < 0.001).

Table 4 shows the children’s tooth brushing habits. In this regard, regular tooth brushing rate of 10-year-old children was significantly lower than those of 8 and 9-year-olds (*p* < 0.001). Furthermore, higher regular tooth brushing rate of girls (26.1%) than the boys’ (22.1%), resulted in a significant difference between the genders (*p* < 0.001).

Table 5 provides data for the children’s dental visits. In terms of dental attendance rate, 10-year-old children had a significantly higher rate than the 8-year-olds (*p* < 0.001). There was no significant difference in the dental visit rates by gender (*p* = 0.901).

## 4. Discussion

PFMs play a key role in the development of a normal occlusion. Therefore, it is very important to protect them for good oral health. Their anatomical characteristics including deep pits, fissures and concavities provide a very favorable environment for plaque accumulation. Parental misunderstanding that PFMs are deciduous and children’s incomplete toothbrushing practices at younger ages in the early PFM eruption stages cause these teeth to develop caries in a short time [16]. Premature PFM loss is commonly observed, especially in the first three years of post-eruptive period when enamel has not been fully mineralized [13,17]. The premature loss of these teeth may cause midline deviation, diastema formations, super eruption of the antagonists, and unilateral mastication resulting in malocclusion [5]. The purpose of the study was to evaluate the PFM status of the children aged 7–10 years in the post-eruptive first three years. The study also compared DMFT scores of PFMs and children’s approaches to oral and dental health regarding age and gender.

The DMFT/dmft index is the most common epidemiological scale used to determine dental health status [18]. In order to prepare and implement the necessary healthcare and treatment programs, it is important to periodically control the changes in these scales through the epidemiological studies to be carried out with different age groups and in different regions. Therefore, using the DMFT index, the present study assessed PFMs of 11,457 children aged 7–10 years between 2015 and 2016 in Erzurum, which was reported to have a high ECC rate. Although the results of this study were not nationwide, they had a high level of reliability and validity because 11,457 children were screened in Erzurum, where a population of 65,136 children aged 5–10 were living.

It has been determined that 15.9% of the PFMs of children in the 7–10 age group in Erzurum had developed caries, and the studies conducted in similar age groups suggested that the rate of caries in PFMs was more than 50% [4,19,20,21]. We realized that the Ministry of Health’s oral and dental health screening programs, training school teachers and students on oral hygiene and 5% NaF varnish applications might have had a positive effect on obtaining lower results in our study [11]. Additionally, we also thought that our epidemiological studies, our activities for parental guidance by determining the dental treatment needs of the children with different ages and our policy of applying not only compulsory but also preventive treatments might have contributed to this outcome. 

Although the present study revealed that girls had a higher tooth brushing rate, their higher rate of decayed and restored PFMs created a statistical difference. It has been stated that higher rates of caries development probability in the girls’ PFMs might have been associated with early puberty and premature eruption, their different dietary and oral hygiene habits, and variations in their quality and quantity of saliva [3,21]. On the contrary, there are also published studies describing higher rates of PFM caries experience in boys [19,22] or in both sexes equally [4,21,23]. It is possible to explain these differences by better personal oral hygiene rather than the gender factor.

It is known that PFMs are fully matured in the first 2 years of post-eruptive stage and they are more vulnerable to caries attacks in this period [24]. This may result in early PFM caries, the need for restoration as the decay progresses with the age, and even the tooth loss. Higher DMFT index scores of PFM associated with age [4,7,19,20] and especially dramatic increases in the caries incidence have been reported in the literature. In their study, Samadani and Ahmad [4] reported that dental caries was observed in at least one PFM of the children aged 9 years at a rate of 67% and this rate increased to 83.5% at the age of 12. Thaker et al. [20] reported that the rate of PFM caries was 37.17% in the 8-year-old group and 62.5% in the 10-year-old group. The present study revealed a PFM caries rate of 7.2% in the 7-year-old group and 23.3% in the 10-year-old group in parallel with the increasing age. Regarding the restoration/caries ratio, in the present study, the ratio of restored/decayed PFMs of the children aged 7 and 8 years was found to be higher than the 9 and 10 years-old group. Caries progression over time and thus need for dental treatment with the emerged pain complaints result in an increased restoration rate with the age [5,6,7].

Previous studies have shown a similar prevalence of bilateral caries occurrence in PFMs [7,20,25,26,27,28]. The present study also revealed a similar PFM caries rate, occurring bilaterally. When the PFMs in the maxilla and mandible were compared, it was determined that the mandibular PFMs were more caries-affected and had a higher restoration rate. Mandibular PFMs are more susceptible to caries mainly due to their complex pit and fissure morphology allowing the accumulation of food residues, earlier eruption time than the maxillary mandibular PFMs and longer exposure to the oral environment [7,28]. In many similar studies, caries incidence was found to be higher in mandibular PFMs than the maxillary molars [7,20,26,28,29].

It is clear that the DMFT score decreases because of the increased level of oral hygiene with regular tooth brushing [23,30]. The present study concluded that the frequency of regular tooth brushing in 10-year-olds was significantly lower than 8- and 9-year-olds. Higher DMFT scores of this age group revealed the effect of tooth brushing on the DMFT score. Similar to our study, Bahannan et al. [30] pointed out that students who brushed regularly (two times a day) had a lower caries prevalence than those who did not brush at all. The same study also concluded that girls had a higher habit of regular tooth brushing and this might be associated with their approaches to personal hygiene and health [30,31]. Although it was observed that girls brushed more regularly than boys in the present study, caries and restoration rates were found to be higher. In addition to the previously mentioned physiological changes in girls, differences in brushing and bacterial plaque removal skills might have led to this situation.

It is possible to take precautions against dental caries and gum diseases through the preventive practices, oral hygiene instructions, and patient motivation, provided after regular dentist visits [4]. Although access to health services is gradually increasing today, many parents think that regular dentist visits are not necessary and they visit dentists only when their children have complaints, such as pain [32,33]. Regular dental visits are in line with the increase in the socioeconomic status of the parents [20]. In the Erzurum province, where households had a little income, [11] only 1.5% of the children in the 7–10 age group had the chance to visit the dentist regularly and this rate was known to be increased in the 10-year-olds. Demirbuga et al. [25] argued that as the age increased in the 6–12 years-old group of children, rate of dental visits also increased because of the severe caries requiring endodontic treatment. It has been stated that this might be due to the complaints of pain increasing with the caries progression or the unwillingness of younger children to visit the dentist because of their dental fear and phobia.

The age groups of 6, 12 and 15 have been determined by the WHO as the target group for caries control [19]. PFMs, which are very important for the development and physiology of the stomatognathic system should be carefully examined and assessed since they fulfill approximately 50% of masticatory function, act as a guide for the eruption of the remaining molars and play a key role in Angle’s Class I occlusion [34]. In the province of Erzurum, DMFT index score of the 7–10-years-old children’s PFMs was found to be lower than similar studies and this result was considered to be very promising for improving these kids’ dental health status in the future. This implies that the oral and dental health applications within the scope of the current health policy are effective and new improvement strategies should be developed for achieving better results. Compulsory dental examination of the age groups targeted by WHO may be a new health policy. However, the low rate of regular dental visits and more than half of the children’s failure to brush their teeth regularly are the issues to be addressed and novel oral and dental health policies may be developed for promoting regular dental checks and tooth brushing.

## 5. Conclusions

In this study, the prevalence of caries in the PFMs of children aged 7–10 years was 15.9% and the mean DMFT was 0.79 ± 1.39. This result showed that PFMs might develop carious lesions and even be lost within three years in the early post-eruptive stage. The current epidemiological study highlights the importance of early evaluation of PFMs in the post-eruptive stage and raising public awareness of oral health in order to reduce DMFT scores.

## Figures and Tables

**Table 1 children-10-00061-t001:** Status of sound, decayed, missing and filled PFMs by gender [n (%)].

	Sound	Decayed	Missing	Filling	Total
Boy	19,574 (85.1%)	3160 (13.7%)	26 (0.1%)	252 (1.1%)	23,012 (100.0%)
Girl	18,389 (80.6%)	4113 (18.0%)	26 (0.1%)	288 (1.3%)	22,816 (100.0%)
Total	37,963 (82.8%)	7273 (15.9%)	52 (0.1%)	540 (1.2%)	45,828 (100.0%)

**Table 2 children-10-00061-t002:** Status of sound, decayed, missing and filled PMFs by age [n (%)].

	Sound	Decayed	Missing	Filling	Total
7	6747 (92.5%)	522 (7.2%)	4 (0.1%)	23 (0.3%)	7296 (100.0%)
8	11,128 (88.0%)	1441 (11.4%)	3 (0.0%)	76 (0.6%)	12,648 (100.0%)
9	10,299 (80.9%)	2247 (17.7%)	16 (0.1%)	166 (1.3%)	12,728 (100.0%)
10	9789 (74.4%)	3063 (23.3%)	29 (0.2%)	275 (2.1%)	13,156 (100.0%)
Total	37,963 (82.8%)	7273 (15.9%)	52 (0.1%)	540 (1.2%)	45,828 (100.0%)

**Table 3 children-10-00061-t003:** Status of sound, decayed, missing and filled PFMs by the tooth number [n (%)].

	Sound	Decayed	Missing	Filling	Total
16	9931 (86.7%)	1445 (12.6%)	11 (0.1%)	70 (0.6%)	11,457 (100.0%)
26	9937 (86.7%)	1435 (12.5%)	11 (0.1%)	74 (0.6%)	11,457 (100.0%)
36	8990 (78.5%)	2246 (19.6%)	15 (0.1%)	206 (1.8%)	11,457 (100.0%)
46	9105 (79.5%)	2147 (18.7%)	15 (0.1%)	190 (1.7%)	11,457 (100.0%)
Total	37,963 (82.8%)	7273 (15.9%)	52 (0.1%)	540 (1.2%)	45,828 (100.0%)

**Table 4 children-10-00061-t004:** Children’s tooth brushing habits [n (%)].

	Not Brushing	Irregular	Once a Day	Regular	Total
7	473 (26.5%)	548 (30.8%)	319 (17.9%)	442 (24.8%)	1782 (100.0%)
8	681 (22.1%)	1067 (34.6%)	551 (17.9%)	782 (25.4%)	3081 (100.0%)
9	701 (22.6%)	1079 (34.7%)	528 (17.0%)	798 (25.7%)	3106 (100.0%)
10	719 (22.4%)	1191 (37.0%)	638 (19.8%)	668 (20.8%)	3216 (100.0%)
Total	2574 (23.0%)	3885 (34.7%)	2036 (18.2%)	2690 (24.1%)	11,185 (100.0%)

**Table 5 children-10-00061-t005:** Children’s dental visits [n (%)].

	Never Gone	Not Gone in the Last Two Years	Gone in the Last Two Years	Going Regularly	Total
7	482 (27.0%)	239 (13.4%)	1028 (57.7%)	33 (1.9%)	1782 (100.0%)
8	694 (22.5%)	620 (20.1%)	1735 (56.3%)	32 (1.0%)	3081 (100.0%)
9	624 (20.1%)	699 (22.5%)	1744 (56.1%)	39 (1.3%)	3106 (100.0%)
10	545 (16.9%)	812 (25.2%)	1797 (55.9%)	62 (1.9%)	3216 (100.0%)
Total	2345 (21.0%)	2370 (21.2%)	6304 (56.4%)	166 (1.5%)	11,185 (100.0%)

## Data Availability

The data that support the findings of this study are available from the corresponding author upon reasonable request.

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
