# Peer review of "Assessment of Permanent First Molars in Children Aged 7 to 10 Years Old"

_children, 2022, doi:10.3390/children10010061_

Round 1

Reviewer 1 Report

The authors need to consider the following points

Please add some sentences to the DMFT system in the introduction, as it is the main tool applies in the study

Materials and methods part cannot be accepted in the current form. Further details to the materials and methods need to be added. Were magnifying glasses used? Was a systematic approach during the examination undertook? Were only cavitated caries registered or also when the color suggested that? Any children were encountered with Syndromes that might involve agenesis of FPM? Which score was recorded when a crown or temporary filling was found?  Among many other details.

Please change the word teeth to (FPM) in all the tables.

Please consider applying a regression analysis of some variables (The regularity of tooth brushing and visiting the dentist for example)

Conclusion should be an answer to the study question. In its current form, conclusion is only a general recommendation.

Author Response

Review 1:

1.Please add some sentences to the DMFT system in the introduction, as it is the main tool applies in the study

“Epidemiological studies play an important role in the distribution and frequency of oral diseases and are considered extremely important for public health, as they form the basis for determining treatment needs. In these studies, there is a need for epidemiological indices that are comparable, reproducible, easily applicable, valid and reliable. [7] In epidemiological studies of dental caries, the DMFT/dmft index comes first among the indexes that meet these features. In the DMFT/dmft index, the general condition of dental health is expressed by the sum of the number of decayed (D/d), missing (M/m) and filled (F/f) teeth.[8]”

2.Materials and methods part cannot be accepted in the current form. Further details to the materials and methods need to be added. Were magnifying glasses used? Was a systematic approach during the examination undertook? Were only cavitated caries registered or also when the color suggested that? Any children were encountered with Syndromes that might involve agenesis of FPM? Which score was recorded when a crown or temporary filling was found?  Among many other details

- No magnifying glass was used. It was evaluated visually under daylight using only a mirror and a probe.

-We do not know whether the absence of these teeth is associated with a syndrome or an ectopic eruption, as we could not obtain panoramic radiographs. however, when we evaluated at our data, we found that children with high overall DMFT/dmft scores had missing teeth. Since a crowned tooth was not encountered, no classification was made for this in our study.

-Added more details to material method section.

“All the FPM have been examined in a systematic approach starting from the upper right FPM and proceeding to the lower right FPM. If a lesion on the tooth surface has an unmistakable cavity, undermined enamel, or a detectable softened floor or wall, the tooth is coded as decayed. In addition, FPM with secondary caries or temporary fillings are also coded as caries. Missing tooth code is used for FPM that have been extracted because of caries. Filled tooth code is used for FPM with permanent restorations and no caries”.

3.Please change the word teeth to (FPM) in all the tables.

"FPM" is used instead of "teeth" in all tables and descriptions.

4.Please consider applying a regression analysis of some variables (The regularity of tooth brushing and visiting the dentist for example)

It was not considered to be included in the study because no significant results were found in the regulation analyses. P value for the regression analysis of tooth brushing and visiting the dentist is 0.19. If you want, we can add it to the findings section.

5.Conclusion should be an answer to the study question. In its current form, conclusion is only a general recommendation.

The conclusion section has been revised.

“When PFMs were evaluated in the same population with a high prevalence of ECC in previous studies, it was found that the DMFT ratio was 17.2% and the DMFT score was 0.79±1.39. This result showed that PFMs might develop carious lesions and even be lost within 1-3 years in the early post-eruptive stage. The current epidemiological study highlights the importance of early evaluation of PFMs in the post-eruptive stage and raising public awareness of oral health in order to reduce DMFT scores.”

Reviewer 2 Report

Assessment of Permanent First Molars in the Early Post-eruptive Stage

Abstract: should contain the aim, results, concrete, and conclusion, not generalities!

Lines 22-23 – should be removed

Introduction – should provide more insight into the necessity of the study and background, as well as a clearly stated aim

Institutional Review Board Statement is missing

Results present only basic percentages of data

Conclusion is not sustained by the results

Lines 251-277 are not filled in.

References are not in journal style

Author Response

Review 2:

1.Abstract: should contain the aim, results, concrete, and conclusion, not generalities!

Abstract section is corrected.

2.Lines 22-23 – should be removed

Lines 22-23 removed.

3.Introduction – should provide more insight into the necessity of the study and background, as well as a clearly stated aim

Background of the study explained. Aim revised.

“The high prevalence of ECC in Erzurum province indicated that PFMs of children in this population were under the risk of carious leisions when the permanent dentition started. In this regard, it was found to be highly significant to examine the PFMs  of children in the same population. The aim of this study is to evaluate the DMFT scores of PFMs in the early post-eruptive stage, within the scope of the United Nations Agenda for 2030 Sustainable Development Goals, thereby raising awareness for the prevention and treatment of permanent tooth decay.”

4.Institutional Review Board Statement is missing

Institutional Review Board Statement is added

5.Results present only basic percentages of data

In the results section, not only the percentages but also the number of data were added.

6.Conclusion is not sustained by the results

The conclusion section has been revised.

“Despite the high prevalence of caries in Erzurum, the DMFT score of DBM teeth in the same population was found to be low. The current epidemiological study emphasized the importance of evaluating the FPM teeth in the early post-emergence period and raising public awareness about oral health in order to reduce the DMFT scores

7.Lines 251-277 are not filled in.

Lines 251-277 removed.

8.References are not in journal style.

References corrected.

Round 2

Reviewer 1 Report

Thanks for responding

Reviewer 2 Report

The manuscript has been improved.